# Tiered manufacturing of pharmaceuticals as a commercial determinant of health: Implications for medicine quality and equity

Jean Christophe Rusatira[1]*, Eishita Pal[1], Jean Berchmans Uwimana[1], Ayesha Khan[1], Manuela Dorado Novoa[1], Shankar Suryanarayanan[2], Lauren Schoukroun-Barnes[3], Christopher Peterson[3], Henry Michtalik[1], Anthony Bakenga Kapeta[4], David Mukanga[5], Saifuddin Ahmed[1], Murray Lumpkin[5], Charles Preston[5]

1 Department of Population, Family and Reproductive Health, Johns Hopkins Bloomberg School of Public Health, Johns Hopkins University, Baltimore, Maryland, United States of America, 2 Operations Transformation LLC, Chicago, Illinois, United States of America, 3 Latham BioPharm Group (LBG), Elkridge, Maryland, United States of America, 4 African Union Development Agency - New Partnership for Africa's Development (AUDA-NEPAD), Midrand, South Africa, 5 Gates Foundation, Seattle, Washington, United States of America

* jrusati1@jh.edu

## Abstract

Tiered manufacturing, the practice of adapting different levels of pharmaceutical production standards to low- and middle-income countries (LMICs) versus high-income countries, has not been investigated as a commercial determinant of health (CDoH), defined by the World Health Organization as a private sector activity affecting public health. This paper examines this practice and its implications for medicine quality, universal health coverage, and global health equity. Guided by the CDoH framework, we conducted semi-structured interviews with 31 mostly India-based experts in pharmaceutical development, manufacturing, procurement, and regulation between May and July 2025. Purposive sampling ensured variation in geography, organizational type, and role. Transcripts were coded deductively using CDoH domains and inductively for emergent themes. Respondents consistently described "tiered manufacturing" — using different production standards by market. Medicines for countries regulated by Stringent Regulatory Authorities (SRA) (now called WHO Listed Authorities (WLA)) adhere to international good manufacturing practices (GMP) through rigorous documentation, stability testing, impurity profiling, and advanced infrastructure. In contrast, products supplied to LMICs are often produced with uncertain GMP compliance, cheaper raw materials, weaker documentation, and less stringent oversight, leading to variable quality and shortened shelf life. Commercial drivers were key determinants. Proposed solutions include improved mechanisms to verify versions of products being received, reliance on WHO prequalification or WLA approvals, procurement conditionality and strengthened local regulatory and manufacturing capacity. Tiered manufacturing was widely perceived as a common

**Data availability statement:** All relevant findings are presented in the manuscript text, supported by anonymized verbatim excerpts throughout the Results section. The interview guide and full codebook are provided as Supplementary Appendices 1 and 2. Because this is a qualitative study involving sensitive industry disclosures, and consistent with the ethics approval granted by the Johns Hopkins Bloomberg School of Public Health Institutional Review Board (IRB no. 00000287), full transcripts and interview notes will not be publicly shared to protect participant confidentiality. A restricted-access data-sharing arrangement may be considered on a case-by-case basis for legitimate research purposes. For queries, please contact bsph.irboffice@jhu.edu.

**Funding:** This study was funded by the Gates Foundation. Funder-affiliated authors ML and CP were included based on their expertise in global pharmaceutical regulation and medicines quality in LMICs and met all four ICMJE authorship criteria. To ensure independence, all data collection, coding, and analysis were completed prior to their review, and their contributions were limited to scientific input on manuscript content. The funder had no role in study design, data collection, analysis, interpretation, writing, or the decision to submit for publication. No author received payment from any pharmaceutical company or other agency to write this article. As the corresponding author, I confirm that JCR and EP had full access to all study data, and all authors accept responsibility for the decision to submit.

**Competing interests:** The authors have declared that no competing interests exist.

practice among respondents. Recognition of this practice as a CDoH is essential to strengthen regulatory reliance and enforcement, procurement leverage, and sustainable capacity-building, ensuring quality medicines for all globally.

## Introduction

Access to safe, effective, quality, and affordable medicines is central to universal health coverage (UHC) [1,2]. Yet which medicines reach which populations, and at what standard, depends as much on economic and commercial dynamics as on public health priorities. The concept of commercial determinants of health (CDoH), defined as "the systems, practices, and pathways through which commercial actors influence health" [1,3], has been applied extensively to industries such as tobacco, alcohol, and ultra-processed foods [4], but rarely to pharmaceutical products.

In this paper we investigate a new potential CDoH of particular significance for low- and middle-income countries (LMICs): the tiered manufacturing of pharmaceuticals. We introduce and operationalize the term "tiered manufacturing" to describe the practice whereby pharmaceutical companies apply systematically different production standards, input quality, and quality assurance processes depending on the destination market. While related concepts such as 'dual-track production,' 'market-differentiated manufacturing,' and 'variable GMP compliance' appear in regulatory and industry discourse, no unified term has been established in the published literature. We propose tiered manufacturing as a conceptually precise and policy-relevant term that captures both the deliberate stratification of production standards and its implications for medicine quality across markets [5]. In its narrowest form, the same product, identical in molecule, dose, and formulation, may be manufactured to different standards for different markets. More broadly, firms may direct products to LMICs that would not meet the requirements of stringent regulatory authority (SRA) or WHO Listed Authority (WLA) markets where drug regulation authorities apply strict standards for quality, safety, and efficacy [5]. Tiered manufacturing fits within the CDoH framework in three respects: it is driven by commercial logic; it operates through structural pathways largely invisible to patients and procurers; and its health consequences, variable medicine quality and therapeutic failure, fall disproportionately on LMIC populations.

Tiered manufacturing is shaped by economic pressures, strategic choices, and regulatory environments, which together influence the quality of medicines available across markets. Evidence, although limited, is concerning. Comparative studies show systematic disparities in dissolution, impurity levels, and shelf life between products destined for SRA/WLA and LMIC markets, particularly in hot and humid climates where degradation is accelerated [6,7]. Whistleblower accounts echo these findings, describing companies "taking their greatest liberties in markets where regulation was weakest and the risk of discovery was lowest," as documented in investigative reporting and subsequent analyses of generic drug manufacturing practices [8]. These practices are often concealed, as disclosure may embarrass companies and provoke consumer outrage, particularly for commodities as sensitive as medicines. Their

consequences are profound, contributing to therapeutic failure, toxic exposures, antimicrobial resistance, financial losses, and erosion of trust in health systems [9,10].

Despite these risks, tiered manufacturing has not been explicitly recognized as a CDoH. Prior research has focused mainly on cost and strategy, without situating manufacturing adaptations as a determinant of population health [11–13]. Globally, at least 10% of medical products are estimated to be substandard or falsified, with prevalence at least twice as high in LMICs, according to WHO and recent systematic investigations of anticancer medicines in sub-Saharan Africa [10,14].

By conceptualizing tiered manufacturing as a CDoH, this paper extends the framework beyond tobacco, alcohol, and unhealthy foods and highlights tiered manufacturing of medical products as an important driver of inequities in medicine quality and access to quality medical products in LMICs.

This study draws on qualitative interviews with professionals and experts in pharmaceutical development, regulation, procurement, and manufacturing to examine how production strategies differ between SRA/WLA and LMIC markets. Specifically, we aim to identify the commercial and regulatory drivers of tiered manufacturing, document stakeholders' perceptions of its consequences for medicine quality, patient safety, and universal health coverage, and explore the policy and regulatory implications of these practices. We argue that tiered manufacturing should be recognized as a commercial determinant of health, with major implications for medicine quality, universal health coverage, and global health equity.

## Methods

### Study design and conceptual framework

This qualitative study examined how pharmaceutical companies adapt production strategies to markets with differing regulatory capabilities, focusing on production processes, sourcing of active pharmaceutical ingredients (APIs) and excipients, quality control, and regulatory oversight when producing for SRA/WLA countries versus LMICs. The study was situated within WHO-GMP-certified pharmaceutical manufacturers and related regulatory and procurement organizations that operate across low- and middle-income country markets. Guided by the CDoH framework, defined as "the systems, practices, and pathways through which commercial actors influence health" [1], we analyzed tiered manufacturing as both a response to economic pressures and a commercial strategy with implications for medicine quality. Following Gilmore and colleagues' conceptualization of the CDoH [3], we situated tiered manufacturing at the intersection of economic pressures, strategic choices, and regulatory environments. Interviews explored differences in facilities, inputs, workforce, and documentation across markets, and the potential consequences for medicine quality.

### Sampling and recruitment

Purposive sampling was used to identify respondents with expertise in pharmaceutical manufacturing, regulation, procurement, or supply chains. A maximum variation approach ensured heterogeneity in geography and professional role. Sample selection eligibility criteria were: (1) direct experience in development, sourcing, manufacturing, quality assurance, procurement, or regulation; (2) ability to communicate in English; and (3) provision of informed consent. We first compiled a list of WHO-GMP-certified companies using the Central Drugs Standard Control Organization (CDSCO) database and complemented this with companies and agencies identified through LinkedIn searches and professional networks. Invitation emails were sent to individuals who responded positively to our LinkedIn messages or to referral messages from fellow respondents. Of 203 experts invited, 32 consented; one was excluded after requesting a non-disclosure agreement, leaving 31 completed interviews.

### Data collection

Participants were recruited between April 1 and June 30, 2025, and semi-structured in-depth interviews were conducted from May 1 to July 30, 2025. The interview guide was structured around tiered manufacturing and the CDoH framework.

It included questions on API sourcing, drug product manufacturing, market segmentation between SRA/WLA-regulated and LMIC markets, and the influence of regulatory oversight and procurement on manufacturing decisions (see S1 Text). Probes explicitly addressed sourcing and grading of excipients, supplier selection, and batch-level quality control and release testing for products destined for different markets.

All the interviews were conducted via Zoom (n = 31). Each lasted 45–75 minutes. Interviews and transcription were conducted by JCR, EP, AK and UJB, with one facilitator and one note-taker. All 31 interviews were audio-recorded and transcribed verbatim and detailed notes were taken and validated immediately post-interview. All transcripts were anonymized and stored securely before analysis.

## Data analysis

Analysis combined deductive and inductive approaches [15]. Deductive coding was informed by domains of the CDoH framework while inductive codes were added as new themes emerged. Coding was undertaken in MAXQDA 2022 [16] by three researchers. To establish a shared coding framework, three coders independently coded an initial subset of five transcripts (16% of transcripts), compared code application, reconciled definitions, and refined the codebook through discussion before proceeding. Discrepancies were resolved through consensus, and the agreed codebook was then applied to the remaining transcripts, each of which was double-coded. Any further disagreements were resolved through discussion.

The final code system comprised 51 codes organized across three hierarchical levels, with 1,870 coded segments in total. Thematic saturation was assessed iteratively: coding of new transcripts was reviewed for the emergence of novel codes, and saturation was considered reached when no new codes were generated across the final five transcripts, corroborated by the high frequency of existing codes in later transcripts. The full codebook, including code definitions and frequencies, is provided in S1 Table.

## Researcher reflexivity

The research team comprised public health researchers and epidemiologists with experience in pharmaceutical regulation, global health policy, and supply chain research in LMIC contexts. While this expertise informed study design and interpretation, it may also have introduced assumptions about tiered manufacturing practices. To mitigate this, the interview guide was reviewed by team members with varying levels of industry familiarity, coding was conducted independently by three researchers before reconciliation, and peer debriefing sessions were held throughout analysis to surface and interrogate interpretive positions. Findings are grounded in participants' own words, with verbatim excerpts used in reporting.

## Ethics

The study protocol was reviewed and approved on January 30, 2025, by the Johns Hopkins Bloomberg School of Public Health Institutional Review Board (Baltimore, MD, USA, IRB no. 00000287). All participants provided electronic informed consent prior to participation and were assured confidentiality.

## Results

### Description of respondents

We interviewed 31 experts representing a wide range of roles across the pharmaceutical sector. (Table 1) Most respondents had more than a decade of experience, with several reporting over 20 years in the industry or regulatory practice. Regulatory affairs and quality assurance were the most common areas of expertise, supplemented by experience in research and development, active pharmaceutical ingredient and finished product manufacturing, procurement, and

**Table 1.** Characteristics of respondents (N = 31).

| Respondent Characteristic | n | % |
|---|---|---|
| **Geographic location** | | |
| India | 26 | 83.9 |
| United States of America | 3 | 9.7 |
| Multi-country (India, USA, Canada) | 2 | 6.5 |
| **Working area** | | |
| Regulatory affairs | 19 | 61.3 |
| Quality assurance/control | 5 | 16.1 |
| Manufacturing (incl. R&D, API synthesis) | 3 | 9.7 |
| Commercial/management | 2 | 6.5 |
| Regulatory affairs & quality assurance | 1 | 3.2 |
| Procurement/supply chain | 1 | 3.2 |
| **Years of experience** | | |
| < 15 years | 11 | 35.5 |
| 15-29 years | 13 | 41.9 |
| 30 + years | 7 | 22.6 |
| Total | 31 | 100 |

commercial strategy. Geographically, more respondents were based in India, reflecting its role as a global hub for medicines, but included respondents with experience in the USA, Canada, multi-country regulatory and procurement contexts. This diversity provided perspectives spanning manufacturing, oversight, and supply chains, from API sourcing to market entry and procurement.

### Recognition, input quality, and drivers of tiered manufacturing

***Recognition of tiered manufacturing as a common practice.*** Respondents consistently acknowledged the existence of tiered or "dual-track" systems, although the terminology itself was not widely used. Respondents described companies as maintaining separate production streams: one aligned with SRA/WLA standards, and another for domestic or LMIC markets, which participants characterized as operating under less stringent regulatory oversight. Smaller firms (described as those that have no more than 150 employees) often framed this as part of a growth trajectory, beginning with less stringent production for local or "rest-of-world" (ROW) markets before upgrading to meet SRA/WLA standards. Larger multinationals were considered less likely to sustain differentiation, owing to reputational risks and the impracticality of parallel systems at scale.

One respondent informed, "Yes, we do have separate plants… they have a relatively next level down standards there" (KII10, Quality Assurance and Control, 30 + Yrs, India).

**Disparities in APIs and excipients across markets.** Disparities in input quality were consistently reported between SRA/WLA-regulated and LMIC markets. According to respondents, APIs and excipients destined for the USA and EU typically required GMP certification and full pharmacopeial documentation, whereas those supplied to LMIC markets were reported to be sourced from lower-grade suppliers with minimal certification to reduce costs.

As one respondent explained, *"For regulated market they always purchase pharma grade, USP [United States Pharmacopeia] grade API, Ph. Eur. [European Pharmacopoeia] excipients. While in non-regulated market, they are*

*compromised with the grade of API and excipients so they can take some profit." (KII11, Regulatory Affairs, 12 + yrs, India)*

**Disparities in manufacturing for LMICs versus SRA markets.** Respondents described marked differences between products destined for SRA/WLA-regulated markets and those for LMICs identified as non-SRA or not in the WLA list. Respondents reported that medicines for regulated markets underwent rigorous documentation, testing, and validation supported by advanced 21 CFR-compliant systems, whereas products for LMICs were perceived to be produced under weaker oversight, with equipment that participants said allowed greater scope for data manipulation.

As one respondent explained, *"In regulated markets, 21 CFR systems prevent data manipulation because every step is recorded. In semi-regulated plants, machinery is often a step down, making manipulation easier"* (KII15, Regulatory Affairs, 15 + yrs, India).

Larger firms with established innovator drug brands in SRA/WLA markets were said to maintain higher and more consistent standards, whereas smaller firms targeting less regulated markets prioritized sales over quality assurance.

One respondent elaborated, *"Large companies have multiple facilities and strong brands, and they use higher-quality systems with substantial manufacturing capacity. Small companies, often with only 100–150 employees, produce mainly for small markets and focus on selling products rather than maintaining quality"* (KII24, Regulatory Affairs, 14 + yrs, India).

**Economic drivers.** Economic pressures were the most frequently cited driver of tiered manufacturing. Producing at lower cost for LMICs was described as a survival strategy in competitive generic markets, influencing testing intensity, input sourcing, and automation. Affordability dictated specifications, higher-grade inputs for wealthier markets and cheaper alternatives for others. Though tiered manufacturing was described as occurring in small and large size companies, it was said larger firms with strong reputations tended to uphold higher standards. Competition from low-cost suppliers, further encouraged less rigorous sourcing even when bioequivalence was uncertain.

One respondent noted, *"[Large company x] maintains very high standards… they spend more on quality, so their product will be more costly… some companies do not"* (KII5, Quality Assurance/Control, 17+ yrs, India)." Another explained, "[some] suppliers… make the API so cheap that you have no option than either discontinue selling in US… or follow the herd" (KII12, Procurement transformation, 15 + yrs, India).

**Strategic drivers.** Beyond economics, strategic factors shaped how firms applied SRA/WLA standards. Respondents cited reputational risk, market growth, and logistical efficiency as key considerations. Some companies maintained higher standards in LMICs as a long-term investment in trust, while others pursued short-term gains where oversight was weak. Publicly listed firms were viewed as more risk-averse, whereas privately held ones often sustained differentiated systems. Contract manufacturing for LMICs was seen as particularly vulnerable to data manipulation due to limited oversight.

As one respondent noted, *"For short term they can cheat… but for long term… they always prefer health first, then business"* (KII11, Regulatory Affairs, 12 + yrs, India). Another added, *"In regulated markets… always a tech transfer… for ROW markets… some data can be manipulated at the CMO [contract manufacturing organization] vendor end"* (KII15, Regulatory Affairs, 15 + yrs, India).

**Regulatory drivers.**  Variation in regulatory environments was seen as central to sustaining tiered manufacturing. Strong, harmonized frameworks in regions such as the US, EU, and Japan compel adherence to high standards, whereas weaker or fragmented systems allow faster registration, minimal data, and higher impurity limits. Such gaps enable lower-quality products and, in some cases, document manipulation, reflecting what regulators are willing or able to enforce rather than outright illegality.

As one respondent elaborated, *"Sometimes the same data submitted to the US can also be submitted to Kenya, which may not require as thorough a review" (KII6, Quality Assurance/Control, 15+yrs, India).* Another added, *"Companies are not even performing those tests… they are doing manipulation in the document without performing" (KII19, Regulatory Affairs, 11+yrs, India, US, Canada).*

### Perceived disparities in product quality and safety across markets

***Stability, shelf life, and batch variability.***  Respondents noted that medicines destined for SRA/WLA markets undergo rigorous stability testing, impurity profiling, and bioequivalence studies, ensuring potency and shelf life. By contrast, products supplied to LMICs were often described as skipping long-term stability studies or reducing batch-level testing and often undergo no bioequivalence testing against the established innovator product. Respondents perceived these practices as contributing to variable assay performance, greater impurity levels, and shorter shelf life, particularly in hot and humid climates where degradation is accelerated, though they acknowledged that systematic measurement of these outcomes was rarely conducted by manufacturers themselves.

As one respondent noted, *"Improper training and weak CAPA [Corrective and Preventive Action] during audits often result in out-of-specification batches… automation is helping reduce these risks, but many sites still rely on manual integration which increases variability" (KII17, Regulatory Affairs, 15+yrs, India).* Another added, *"In the US specifically, you have to manufacture based on attribute batch by batch, not commercial batches generally" (KII6, Quality Assurance/Control, 15+yrs, India).*

**Excipient and input safety.**  Beyond APIs, respondents highlighted excipients as a critical dimension of product safety and cost determinants. For regulated markets, excipient vendors were expected to provide detailed certification packages, including nitrosamine and elemental impurity studies. In LMIC markets, however, manufacturers frequently sourced from lower-tier suppliers, with documentation limited to basic certificates of analysis. One respondent characterized system of "A," "B," or "C" category excipients, where poorer markets often received the lowest-quality categories, though this framing reflects one participant's characterization rather than a formally recognized industry classification.

As one respondent explained, *"In regulated markets, every step must meet strict GMP requirements. For smaller companies, maintaining those standards can be difficult, especially for excipients. I have seen the same product supplied to Europe and to African countries, but with different excipient grades—USP or CEP [Certificate of Suitability to the European Pharmacopoeia] grade for regulated markets and lower-grade materials for emerging markets where requirements are less strict." (KII18, Regulatory Affairs, 14+yrs, India).* Another added, *"In ROW markets, inspections often check GMP processes and SOPs, but not the product itself… unlike regulated markets where inspections cover formulation, stability, and product lifecycle" (KII15, Regulatory Affairs, 15+yrs, India).*

**Patient safety implications.**  We also inquired about concerns related to patient safety risks associated with tiered manufacturing. Respondents highlighted that cost-driven procurement decisions often led governments and buyers in

LMICs to accept medicines of compromised quality, and that the absence of simple markers made it difficult to distinguish manufacturers serving regulated versus non-regulated markets.

As one respondent explained, *"Albendazole for deworming children in Africa and Southeast Asia… is of such low quality that it doesn't even dissolve properly. Yet governments procure and use it because it is the only drug available"* (KII1, R&D, 20 + years, India, US). Another added, *"There is no simple parameter to identify if a company produces for regulated or non-regulated markets… but CEP listings and the Orange Book in the US allow us to know who supplies higher-standard medicines"* (KII11, Regulatory Affairs, 12 + yrs, India).

### Addressing the disparities: Solutions and mitigation strategies

*Regulatory harmonization, recognition, and reliance.* Respondents frequently emphasized that fragmented oversight across LMICs weakens quality assurance. Regional initiatives such as the African Medicines Agency (AMA) were described as essential to harmonize standards, reduce duplication, and pool scarce technical and enforcement expertise. Participants also highlighted reliance mechanisms, such as recognition of approvals or inspection outcomes from stringent regulatory authorities (SRAs/WLAs) or WHO prequalification as practical pathways for regulators with limited resources, provided they receive the same product version assessed and inspected by the SRA/WLA or WHO, which in many tiered manufacturing scenarios could be difficult for the regulators in LMICs to determine.

As one respondent explained, *"If LMIC regulators can rely on WHO PQ [prequalification] or SRA approvals, they don't need to repeat everything, recognition can save time and cost"* (KII24, Regulatory Affairs, 14 + yrs, India).

**Leveraging market power and transparency.** Market-shaping was considered a powerful tool to drive higher standards. Respondents recommended pooled procurement and conditional purchasing tied to WHO prequalification or SRA/WLA authorization, even when cheaper alternatives exist. Donors and major procurement agencies were seen as best positioned to enforce these requirements. Transparency measures, such as independent audits, public inspection reports, and patient reporting tools like hotlines or mobile apps, were also emphasized to enhance accountability and detect poor-quality medicines more quickly.

As one respondent noted, *"If procurement agencies clearly state that only WHO-prequalified or SRA-approved products will be purchased, then manufacturers will comply. Market leverage works faster than regulation in many cases"* (KII11, Regulatory Affairs, 12 + yrs, India). Another added, *"Public disclosure of inspections and third-party audits can make a big difference, if a company knows their practices will be visible, they will think twice before cutting corners"* (KII3, Quality Assurance/Control, 17 + yrs, India).

**Building capacity, harnessing technology, and ensuring sustainability.** Respondents emphasized that capacity-building must extend beyond regulators to the wider manufacturing and health workforce. Training factory workers in GMP, continuous professional development for QA staff, and refresher courses for inspectors were viewed as essential for sustaining quality improvements. Technology, such as electronic batch records, track-and-trace systems, and automation, was seen as critical to prevent data manipulation, while sustainable financing beyond donor start-up support was deemed vital to maintain laboratory and digital infrastructure.

As one respondent summarized, *"GMP training should not just be for regulators; factory workers also need to know why each step matters, otherwise the same mistakes repeat"* (KII3, Quality Assurance/Control, 17 + yrs, India).

Across interviews, respondents described a clear contrast between products destined for SRA/WLA-regulated high-income markets and those supplied to LMICs. For high-income countries, respondents perceived that companies emphasized stringent GMP compliance, higher-grade APIs and excipients, extensive stability and bioequivalence testing, and 21 CFR-compliant digital systems that limit data manipulation. In LMIC markets, by contrast, respondents reported greater use of lower-grade inputs, reduced testing, and more manual or fragmented documentation, facilitated by weaker or less harmonized regulatory oversight. Several respondents noted that this tiered approach allows companies to offer lower-priced products that support coverage in resource-constrained settings, but often at the cost of shorter shelf life, greater batch variability, and higher risks of substandard quality.

## Discussion

This study provides qualitative evidence that tiered manufacturing is perceived as a widespread practice, shaped by economic incentives, strategic decisions, and uneven regulatory oversight. Respondents consistently described differences between products destined for SRA/WLA-regulated markets and those for LMICs, with the latter often made using lower-grade APIs and excipients, weaker documentation, and less robust quality assurance oversight. Although not always unlawful, such practices exploit differences in regulatory requirements and enforcement capabilities and raise fundamental questions of health equity and patient safety.

From a universal health coverage perspective, these findings suggest that tiered manufacturing may both support and undermine health coverage in LMICs. Lower-cost production strategies help governments and donors to procure larger volumes of medicines, potentially expanding nominal coverage, yet respondents consistently linked these same strategies to weaker quality systems and higher risks of substandard products reaching patients. Thus, tiered manufacturing functions as a commercial determinant of health that shifts the coverage–quality trade-off onto populations in LMICs, raising critical equity and patient-safety concerns.

Within the CDoH framework, tiered manufacturing exemplifies how commercial actors influence health through market practices rather than individual consumer choices. Distinctively, it exploits the gap between what a product could be and what is permitted in markets with weaker oversight, making regulatory environments both the enabler and the potential solution [1–3]. These determinants reflect the interplay of economic pressures, strategic choices, and regulatory environments [17]. Firms adjust manufacturing standards to market income levels and oversight strength, aligning with commercial incentives but undermining universal health coverage and equity. Budget constraints in LMICs drive procurement of the lowest-cost tenders, encouraging cost-cutting on APIs, excipients, and quality systems [2,18–20]. Smaller firms use LMIC production as a growth path before meeting SRA/WLA or PQ standards [11,21], while most larger companies especially those manufacturing innovator products tend to balance reputational risks with profit. But for firms producing mainly for LMICs, reputational risk was seen as minimal given the perceived limited testing and oversight capacity of target NRAs. Weak and fragmented regulation sustains these disparities, enabling lower data requirements, higher impurity thresholds, and documentation manipulation [20]. These patterns show how commercial logics shift risks to disadvantaged populations while concentrating benefits where returns are greatest [2,22].

The consequences are stark. Substandard and falsified oxytocin, antibiotics, albendazole, tuberculosis and oncology products have been documented in LMICs [6,9,23–25].

Recent diethylene glycol poisonings in The Gambia, Uzbekistan, and Indonesia, underscore the human cost of weak oversight [26]. Such disparities contribute to therapeutic failure, antimicrobial resistance, toxicity, and financial losses, while eroding trust in health systems. They may also undermine reliance, as sameness of products cannot be assumed, risking reinforcement of inequities [27].

Solutions require joint regulatory and procurement action. Regional harmonization efforts such as the AMA and reliance on WHO prequalification or SRA/WLA authorizations can strengthen capacity and reduce duplication [27]. Procurement bodies and donors can shape markets through pooled purchasing, conditional contracts, and requirements for WHO

prequalified or WLA/SRA-approved products, leveraging purchasing power to align incentives [18]. The hope is that with time the newly opened AMA will be accredited by WHO as a WLA and thus would be then a strongly performing regional reference authority to help address this problem. Greater transparency, through inspection reports, independent audits, and patient reporting mechanisms can increase accountability. Investment in capacity training and digital technologies such as electronic batch records and track-and-trace systems will be essential for sustainability. These strategies echo prior evidence that global incentives linked to uniform standards can shift markets towards quality [28].

This study has limitations. Purposive sampling and restriction to English-speaking participants mostly based in India may have introduced bias. The sensitivity of the topic could have discouraged disclosure, although anonymity of responses suggests this was limited. Strengths include the breadth of perspectives across industry, regulation, and procurement, and the use of a well-established CDoH framework. In addition, information saturation was achieved, with no new themes emerging in the final interviews, supporting the adequacy of the sample. To enhance rigor, we followed established guidance for qualitative validity and reporting [29]. We also acknowledge that tiered manufacturing is a term we introduce and operationalize in this paper. While the underlying practices have been described in regulatory reports and investigative accounts, a unified term has been absent from the academic literature. We propose this term as a contribution to the field and recognize that its acceptance will depend on validation through future quantitative and regulatory research.

## Conclusion

This study shows that tiered manufacturing is neither marginal nor incidental, but a systemic practice shaped by economic incentives, strategic decisions, and uneven regulatory environments. By documenting how companies adapt inputs, processes, and oversight depending on destination markets, we highlight tiered manufacturing as a commercial determinant of health with implications for medicine quality, universal health coverage, and global health equity.

Although lower-cost production may support affordability in resource-constrained settings, these savings often come at the expense of rigorous quality controls, with consequences for patient safety and trust in health systems. Access to a medicine is not access unless it is access to a quality version. Recognition of tiered manufacturing as a determinant of health is therefore essential to framing responses. Regulatory harmonization and reliance on stringent approvals or WHO prequalification are critical to ensuring that LMIC markets receive the same product versions authorized and inspected by WHO or SRA/WLA regulators. Market-shaping procurement policies that prioritize quality over cost, alongside investment in surveillance, transparency, and manufacturing capacity, will be essential for sustainability.

Global actors, including WHO, procurement agencies, and donors, should explicitly integrate GMP oversight of tiered manufacturing into strategies to combat SF medical products and quantify the scale and impact of tiered manufacturing and evaluate the effectiveness of harmonization and procurement reforms. Only through coordinated global and national action can all populations be assured access to medicines that are both affordable and of consistent quality.

## Supporting information

**S1 Table. Code system.**
(DOCX)

**S1 Text. Interview guide.**
(DOCX)

## Author contributions

**Conceptualization:** Jean Christophe Rusatira, Shankar Suryanarayanan, Lauren SCHOUKROUN-BARNES, Christopher Peterson, Henry Michtalik, Anthony Bakenga Kapeta, David Mukanga, Saifuddin Ahmed, Charles Preston.

**Data curation:** Jean Christophe Rusatira, Eishita Pal, Ayesha Khan.

**Formal analysis:** Jean Christophe Rusatira, Eishita Pal, Jean Berchmans Uwimana, Ayesha Khan, Manuela Dorado Novoa, Shankar Suryanarayanan, Lauren SCHOUKROUN-BARNES, Saifuddin Ahmed, Charles Preston.

**Investigation:** Jean Christophe Rusatira.

**Methodology:** Jean Christophe Rusatira, Eishita Pal, Jean Berchmans Uwimana, Ayesha Khan, Manuela Dorado Novoa, Shankar Suryanarayanan, Christopher Peterson, Henry Michtalik, David Mukanga, Saifuddin Ahmed, Murray Lumpkin, Charles Preston.

**Resources:** Jean Christophe Rusatira.

**Supervision:** Saifuddin Ahmed, Murray Lumpkin, Charles Preston.

**Validation:** Jean Christophe Rusatira, Eishita Pal, Jean Berchmans Uwimana, Anthony Bakenga Kapeta, David Mukanga, Saifuddin Ahmed, Murray Lumpkin, Charles Preston.

**Visualization:** Jean Christophe Rusatira.

**Writing – original draft:** Jean Christophe Rusatira.

**Writing – review & editing:** Jean Christophe Rusatira, Eishita Pal, Jean Berchmans Uwimana, Ayesha Khan, Manuela Dorado Novoa, Shankar Suryanarayanan, Lauren SCHOUKROUN-BARNES, Christopher Peterson, Henry Michtalik, Anthony Bakenga Kapeta, David Mukanga, Saifuddin Ahmed, Murray Lumpkin, Charles Preston.

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
