## [Decision Letter · Decision Letter 0]

3 Mar 2026

PGPH-D-25-03840

Tiered manufacturing of pharmaceuticals as a commercial determinant of health: implications for medicine quality and equity

Dear Dr. Rusatira,

Thank you for submitting your manuscript to PLOS Global Public Health. After careful consideration, we feel that it has merit but does not fully meet PLOS Global Public Health’s publication criteria as it currently stands. Therefore, we invite you to submit a revised version of the manuscript that addresses the points raised during the review process.

The manuscript has been evaluated by three reviewers, and their comments are available below.

The reviewers have raised a number of major concerns. Reviewers have highlighted issues related to the background for the study, sampling and methodological detail, as well as the interpretation of findings.

Could you please carefully revise the manuscript to address all comments raised?

We look forward to receiving your revised manuscript.

Kind regards,

Ilse Bloom

Staff Editor

Journal Requirements:

According to our Data Policy, the contact point must not be an author on the manuscript and must be an institutional contact, ideally not an individual. Please revise your data statement to a non-author institutional point of contact, such as a data access or ethics committee, and send this to us via return email. Please also include contact information for the third-party organization, and please include the full citation of where the data can be found.

Additional Editor Comments (if provided):

Reviewers' comments:

Reviewer's Responses to Questions

**Comments to the Author**

1. Does this manuscript meet PLOS Global Public Health’s publication criteria? Is the manuscript technically sound, and do the data support the conclusions? The manuscript must describe methodologically and ethically rigorous research with conclusions that are appropriately drawn based on the data presented.

Reviewer #1: Yes

Reviewer #2: Partly

Reviewer #3: Yes

2. Has the statistical analysis been performed appropriately and rigorously?

Reviewer #1: Yes

Reviewer #2: No

Reviewer #3: N/A

3. Have the authors made all data underlying the findings in their manuscript fully available (please refer to the Data Availability Statement at the start of the manuscript PDF file)?

Reviewer #1: Yes

Reviewer #2: No

Reviewer #3: No

4. Is the manuscript presented in an intelligible fashion and written in standard English?

Reviewer #1: Yes

Reviewer #2: No

Reviewer #3: Yes

5. Review Comments to the Author

Reviewer #1: Comments

Title: Tiered manufacturing of pharmaceuticals as a commercial determinant of health: implications for medicine quality and equity

• concise, clear and specific title

Abstract: well written; except the study setting (country) should be mentioned in the method part.

Introduction

The introduction part tried to comprehensively present the research problem or gaps and well justified why the research is needed. However, some points mentioned below should be modified to improve the quality.

i. Some information in this section is not from latest literatures. Therefore, it is better if latest literatures are used. Example, try to replace reference ‘Eban K. Dirty medicine. Fortune. 2013’ by other latest reference(s).

ii. Check for secondary citation or indirect citation. Example statement from reference number 15 (Globally, at least 10% of medicines are estimated to be substandard or falsified, with prevalence at least twice as high in LMICs) is indirect citation.

Method

Study design and conceptual framework: Adding brief description about the selected pharmaceutical companies for the study and their selection technique can make the study setting clearer for the readers.

Sampling and recruitment: Was the invited 203 experts for interview the total number of potential study participants or from how many potential study participants this number chosen?

Data collection: According to the study objective, this study focus also includes sourcing of pharmaceutical excipients and quality control. However, their inclusion in questions for data collection is missing or not mentioned. Therefore, it needs clarification.

Ethics: Clarify when the study protocol approval confirmed and present ethical clearance code.

Results

Description of respondents

The distribution of respondents across their geographic location and their working area (manufacturing, regulatory affairs and quality assurance, procurement or supply chain) should be presented. These variables can clarify the respondents’ background, and their diversities can directly influence their responses.

Discussion

The main concepts of the study, tiered manufacturing, should be explained by comparing the similarities and differences for high-income countries and LMICs using key findings. Some key findings like the potential use of the tiered manufacturing in maintaining the health coverage of LMICs in relation to its impacts should be well addressed or discussed. It needs reconsideration.

Conclusion

Well written. It is drafted from the study findings and well presented.

Reviewer #2: Dear Authors,

Thank you for submitting this important study titled “Tiered Pharmaceutical Manufacturing as a Commercial Determinant of Health.”. The manuscript addresses a critical topic related to pharmaceutical manufacturing practices and global health equity. The study has strong potential; however, several issues must be addressed to improve the scientific rigor, clarity, and credibility of the findings.

Below are detailed comments structured for revision.

Major Comments

1. Clarification of Research Objectives and Hypotheses

The manuscript outlines the importance of tiered manufacturing but does not clearly define the specific research questions or hypotheses guiding the study.

Suggested Improvement:

Please explicitly state the research questions in the introduction and clarify whether the study aims to:

• Identify drivers of tiered manufacturing,

• Document perceived consequences,

• Evaluate policy implications.

2. Sampling Structure and Representativeness

The participant pool appears unevenly distributed geographically, with strong representation from specific regulatory environments. This limits the ability to interpret findings as globally representative.

Suggested Improvement:

• Please include a detailed participant table describing:

• Country or region of expertise

• Sector (regulator, industry, procurement, academia)

• Years of experience

• Professional role.

• Discuss potential selection bias and limitations in sampling.

3. Methodological Transparency

The manuscript briefly mentions coding and thematic analysis but lacks sufficient detail regarding analytical procedures.

Suggested Improvement:

Please provide:

• Interview guide (supplementary material)

• Codebook

• Description of coding workflow

• Explanation of coder agreement process

• Details on how thematic saturation was determined.

4. Researcher Reflexivity

The manuscript does not adequately describe the researchers’ roles and potential influence on the study. Reflexivity is an important element of qualitative research quality assessment.

Suggested Improvement:

• Add a reflexivity statement including:

• Researchers’ professional backgrounds

• Potential sources of bias

• Steps taken to minimize bias.

5. Interpretation of Findings

Some findings are presented as established patterns rather than perceptions expressed by interview participants. Qualitative studies capture expert experiences but cannot establish prevalence.

Suggested Improvement:

Please revise wording to clarify that findings represent expert perspectives rather than confirmed global trends.

6. Evidence Supporting Key Claims

Certain statements about pharmaceutical quality differences and manufacturing practices rely mainly on interview testimony. These claims are sensitive and require strong evidence or careful framing.

Suggested Improvement:

• Provide supporting empirical literature where possible.

• Include additional anonymized quotes supporting key themes.

• Clearly distinguish between perceptions and documented evidence.

7. Data Availability and Transparency

The manuscript states that interview transcripts will not be publicly shared. This may conflict with journal data-sharing policies.

Suggested Improvement:

• Consider providing:

• De-identified excerpts

• Coding matrices

• Codebook

• A restricted-access data-sharing mechanism.

8. Conflict of Interest and Funding Disclosure

The role of funder-affiliated authors requires clearer explanation. Transparency regarding funding influence is essential for research credibility.

Suggested Improvement:

• Clarify Whether the funder had involvement in analysis or interpretation.

• How independence of the research process was ensured.

Reviewer #3: Manuscript:

Tiered manufacturing of pharmaceuticals as a commercial determinant of health:

implications for medicine quality and equity

General: The manuscript assesses an important concept that contributes poor quality medicines circulation in low and middle income countries(LMICs)

Major comments:

1. Does the term "tiered manufacturing" appropriate to describe the challenges the manuscript focused?. It is not described in the literature before and thus it might be needed to emphasize on working definition of it and promoting he term to be accepted in the field.

2, It will be better if the study tool is included as supplementary file in the manuscript.

3. The relation ship between the commercial determinant of

health (CDoH) and tiered manufacturing is not clearly shown in the manuscript.

Minor:

Some references are not per the PLOS Global Public health guideline like reference 2.

6. PLOS authors have the option to publish the peer review history of their article (what does this mean?). If published, this will include your full peer review and any attached files.

**Do you want your identity to be public for this peer review?** For information about this choice, including consent withdrawal, please see our Privacy Policy.

Reviewer #1: No

Reviewer #2: **Yes:** Rashed Ahmed

Reviewer #3: No

Figure Resubmissions:

---

## [Decision Letter · Decision Letter 1]

15 May 2026

Tiered manufacturing of pharmaceuticals as a commercial determinant of health: implications for medicine quality and equity

PGPH-D-25-03840R1

Dear Rusatira,

We are pleased to inform you that your manuscript 'Tiered manufacturing of pharmaceuticals as a commercial determinant of health: implications for medicine quality and equity' has been provisionally accepted for publication in PLOS Global Public Health.

Best regards,

Julia Robinson

Executive Editor

Reviewer Comments (if any, and for reference):

Reviewer's Responses to Questions

**Comments to the Author**

1. If the authors have adequately addressed your comments raised in a previous round of review and you feel that this manuscript is now acceptable for publication, you may indicate that here to bypass the “Comments to the Author” section, enter your conflict of interest statement in the “Confidential to Editor” section, and submit your "Accept" recommendation.

Reviewer #3: All comments have been addressed

2. Does this manuscript meet PLOS Global Public Health’s publication criteria? Is the manuscript technically sound, and do the data support the conclusions? The manuscript must describe methodologically and ethically rigorous research with conclusions that are appropriately drawn based on the data presented.

Reviewer #3: Yes

3. Has the statistical analysis been performed appropriately and rigorously?

Reviewer #3: N/A

4. Have the authors made all data underlying the findings in their manuscript fully available (please refer to the Data Availability Statement at the start of the manuscript PDF file)?

Reviewer #3: Yes

5. Is the manuscript presented in an intelligible fashion and written in standard English?

Reviewer #3: Yes

6. Review Comments to the Author

Reviewer #3: The authors had addressed the comments forwarded in proper scientific way.

7. PLOS authors have the option to publish the peer review history of their article (what does this mean?). If published, this will include your full peer review and any attached files.

**Do you want your identity to be public for this peer review?** For information about this choice, including consent withdrawal, please see our Privacy Policy.

Reviewer #3: No
